# FERARI and cargo adaptors coordinate cargo flow through sorting endosomes

Jachen A. Solinger ®[1], Harun-Or Rashid[1] & Anne Spang ®[1]✉

Cellular organization, compartmentalization and cell-to-cell communication are crucially dependent on endosomal pathways. Sorting endosomes provide a transit point for various trafficking pathways and decide the fate of proteins: recycling, secretion or degradation. FERARI (Factors for Endosome Recycling and Rab Interactions) play a key role in shaping these compartments and coordinate Rab GTPase function with membrane fusion and fission of vesicles through a kiss-and-run mechanism. Here, we show that FERARI also mediate kiss-and-run of Rab5-positive vesicles with sorting endosomes. During these encounters, cargo flows from Rab5-positive vesicles into sorting endosomes and from there in Rab11-positive vesicles. Cargo flow from sorting endosomes into Rab11 structures relies on the cargo adaptor SNX6, while cargo retention in the Rab11 compartment is dependent on AP1. The available cargo amount appears to regulate the duration of kisses. We propose that FERARI, together with cargo adaptors, coordinate the vectorial flow of cargo through sorting endosomes.

Cells are constantly interacting and exchanging materials and signals with their surroundings. To accomplish this, they need efficient machineries for uptake and recycling of proteins and solutes. While the uptake of cargo through endocytosis has been extensively studied and is quite well understood, the recycling and sorting part of the cycle is less clear[1,2]. Many factors involved in endosomal sorting and recycling have been described[3], and microscopic analyses revealed a fascinating and complex network of dynamic tubules, where these processes take place[4]. Current models concentrate on sorting events occurring directly at Rab5-positive structures[5]. Sorting nexins (SNXs) with membrane tubulation activities form tubules, where adaptor proteins are recruited and attract cargo. These transport carriers acquire the appropriate Rab GTPase (e.g., Rab11) and pinch off to form vesicles with defined cargoes, which can be transported to their final destination. Theses transport routes use retromer and retriever complexes for the transport for a large variety of cargoes[6]. While this model addresses many features of cargo sorting and recycling, it falls short to explain the presence of large tubular recycling networks or how low binding affinities of cargo adaptors can ensure efficient separation and sorting of cargoes.

We recently described an additional mechanism by which cargo sorting and recycling occurs at sorting endosomes. In this process, the FERARI tethering complex promotes a kiss-and-run interaction between Rab11-positive recycling vesicles and tubular SNX1-sorting endosomes[7]. We inferred that during the kiss, cargo would pass from the SNX1 to the Rab11 compartment. We found that recycling of human transferrin receptor (hTfR) and GLUT1 in the *C. elegans* intestine and transferrin (Tfn) in mammalian cells are dependent on FERARI-mediated kiss-and-run, and that interfering with FERARI function greatly reduced recycling of these cargoes[7]. FERARI contains proteins for tethering (Rab11FIP5, Rabenosyn-5), SNARE interactions (VPS45), binding to SNXs (VIPAS39), scaffolding and protein-protein interactions (ANK1) as well as membrane tubule stabilization and pinching (EHD1)[7]. Thus, the FERARI tether uniquely combines membrane fusion with membrane fission activity. Most of the FERARI components have been shown previously to be involved in endocytic traffic and also in processes independent of FERARI. For example, Rabenosyn-5 and Vps45 are also effectors of Rab5 on PI3P-positive endosomes[8]. Nevertheless, knockout or knockdown of individual FERARI members all abolished kiss-and-run of Rab11 vesicles with SNX1-sorting endosomes[7]. The integration of all these factors for recycling into one machinery allows for a tightly regulated mechanism of tethering, docking, cargo exchange and fission of vesicles, which serve as a basis for the observed kiss-and-run behavior of Rab11

[1]Biozentrum, University of Basel, Spitalstrasse 41, 4056 Basel, Switzerland. ✉e-mail: anne.spang@unibas.ch

recycling vesicles. However, more studies are needed to fully understand the process of cargo sorting in endosomes, in particular by kiss-and-run.

In this study, we unexpectedly find that Rab5 and Rab10-positive endosomes also undergo FERARI-mediated kiss-and-run at SNX1 endosomes. The observed kinetics are similar to those observed for Rab11-positive vesicles. Furthermore, cargo flows from Rab5-positive

endosomes into the SNX1-sorting compartment and from there into Rab11 recycling endosomes. We surmise that Hrs/ESCRT-0 retains cargoes in Rab5 endosomes destined for the degradative pathway, while the rest of the cargoes would be free to reach the SNX1-positive tubules by FERARI-mediated kiss-and-run, from which they can reach recycling endosomes. Directionality of cargo flow into and through sorting endosomes into recycling endosomes would be achieved by

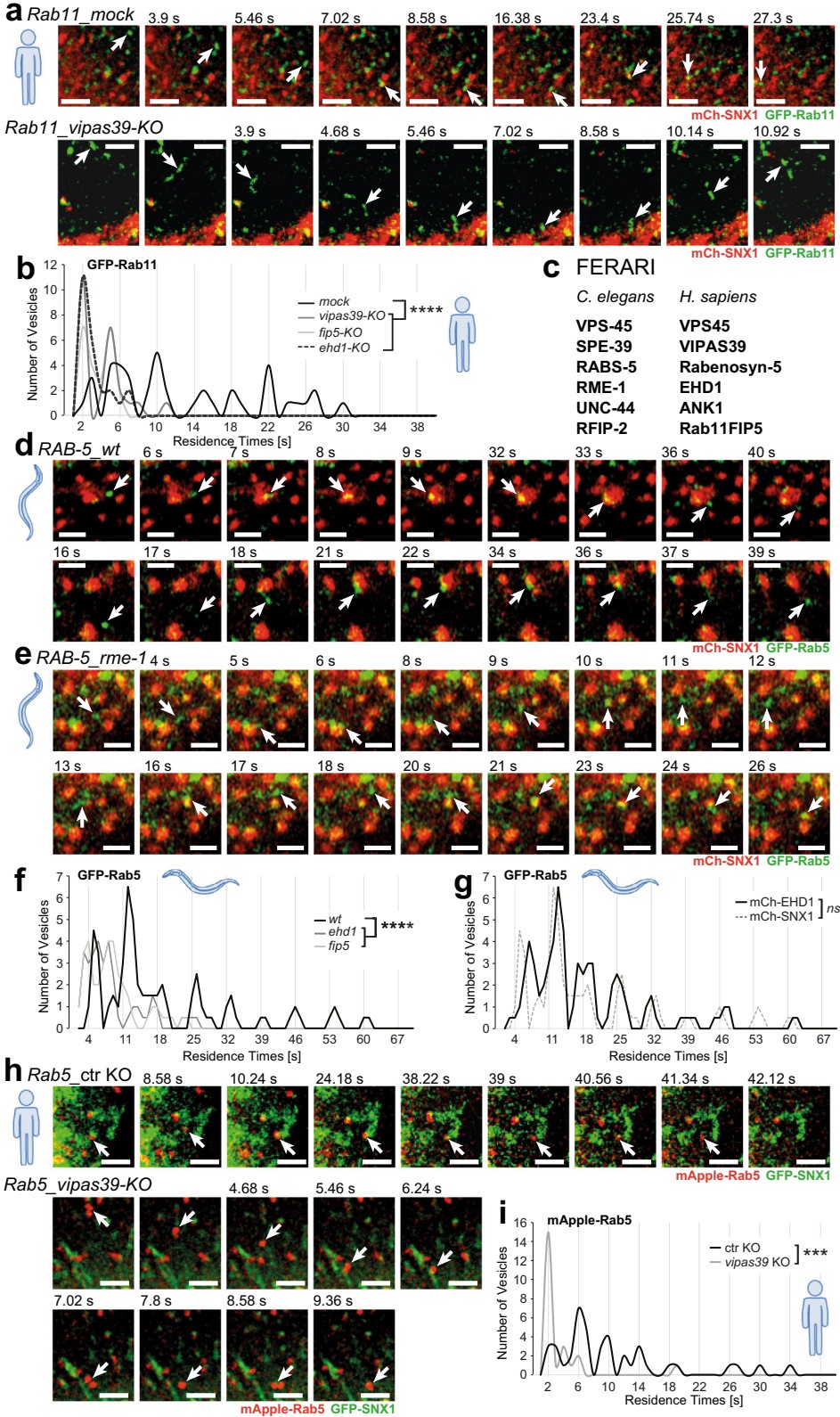

**Fig. 1 | Rab5 and Rab11 vesicles exhibit kiss-and-run behavior dependent on FERARI in both worms and HeLa cells. a** Rab11 vesicles perform kiss-and-run dependent on FERARI. Movie stills showing kiss-and-run of endogenously GFP-tagged Rab11 vesicles (arrow) in ctr KO ($n = 41$) and *vipas39-KO* ($n = 25$) HeLa cells (see also Supplementary Movie 1). Scale bar: 2 μm. **b** Rab11 vesicles dock on SNX1 networks with residence times in distinct intervals. Groups of vesicles with similar residence times appear as peaks (see also Supplementary Fig. 2b for single-vesicle graph, *mock*: $n = 41$, *vipas39-KO*: $n = 25$). This pattern is abolished in *vipas39*-KO cells (see Supplementary Fig. 2a, b for *ehd1-KO* and *rab11fip5-KO* additional data). One-way multiple comparison ANOVA test P-values: *mock-vipas39-KO P* = 1.98E-09, *mock-ehd1-KO P* = 1.6E-10, *mock-rab11fip5-KO P* = 1.58E-09. **c** List of FERARI genes in *C. elegans* and *H. sapiens*. The human nomenclature is used throughout the manuscript. The model organism is indicated by a human or worm icon next to the data. **d** Movie stills showing kiss-and-run of Rab5 vesicles (arrow) in *wild-type* worms ($n = 53$). Scale bar: 2 μm. **e** Movie stills showing kiss-and-run of Rab5 vesicles (arrow) in *ehd1(RNAi)* worms ($n = 33$) (see also Supplementary Movie 3). Scale bar: 2 μm. **f** Rab5 vesicles dock on SNX1 networks with residence times in distinct intervals (see also Supplementary Fig. 3a for single-vesicle graph, $n = 53$). This pattern is abolished in *ehd1* ($n = 33$) and *fip5* ($n = 35$) knock-downs (see Supplementary Fig. 3a for single-vesicle plots). One-way multiple comparison ANOVA test P-values: *mock-ehd1 P* = 4.63E-05, *mock-fip5 P* = 3.83E-05. **g** Residence times intervals in Rab5 vesicles docking to mCherry-EHD1 compartments (*wild-type* from **f** as comparison, $n = 63$) (see Supplementary Fig. 3a). Unpaired two-tailed *t*-test: $P = 0.8467$. **h** Rab5 exhibits kiss-and-run behavior in HeLa cells stably expressing mApple-Rab5 and transiently expressing GFP-SNX1. Movie stills show representative vesicles for ctr KO ($n = 41$) and *vipas39*-KO ($n = 23$) cells (arrow) (see also Supplementary Movie 4). Scale bars: 2 μm. **i** Binning of vesicles from (**h**) reveals distinct intervals of residence times for Rab5 vesicles (*mock*: $n = 41$, *vipas39-KO*: $n = 23$). This pattern is abolished in *vipas39-KO* cells (see Supplementary Fig. 3b for single-vesicle plots). Unpaired two-tailed *t*-test: $P = 0.000118$.

cargo adaptors and adaptor complexes such as SNX6 and AP1, respectively. Moreover, we find that cargo availability in SNX1-sorting endosomes contributes to the duration of the kiss between sorting and recycling endosomes. We propose that interactions between cargoes and their adaptors control their vectorial flow from early to recycling endosomes. This additional, conserved pathway may increase the robustness and capacity of sorting and recycling of cargoes to the plasma membrane.

## Results

### Kiss-and-run between Rab11 and SNX1 compartments is conserved in mammalian cells

While we have previously shown that the FERARI tethering complex is conserved in metazoan and our data were also consistent with a functional conservation, we had not shown that FERARI-dependent kiss-and-run of Rab11 vesicles on SNX1-sorting endosomes occurred[7]. Therefore, we imaged HeLa cells with endogenously tagged GFP-Rab11 (Supplementary Fig. 1b) expressing mCherry-SNX1. Indeed, we observed kiss-and-run events with a similar quantal behavior as in *C. elegans*[7] (Fig. 1a, b and Supplementary Movie 1), with the sole difference that the residence times increased with a 4-s periodicity, compared to 7 s in *C. elegans*. The basis of this difference in the periodicity remains unclear. We speculate that 4 or 7 s intervals are a reflection of the process of fusion pore opening, cargo flux and membrane scission, and that the periodicity is an indication of an unsuccessful fission event(s) followed again by fusion pore opening cargo flux and another fission attempt. Like in *C. elegans*, these kiss-and-run events required FERARI function, as individual knockout lines of FERARI members abolished kiss-and-run between Rab11 vesicles and SNX1-sorting compartments (Fig. 1a, b; Supplementary Fig. 2a, b and Supplementary Movie 1). We conclude that the function of FERARI is conserved from *C. elegans* to mammals. Besides the kiss-and-run of Rab11 with SNX1 compartments, we also observed sometimes the previously described pinching-off of Rab11-positive tubules in *C. elegans* and mammalian cells (Supplementary Fig. 2c, d and Supplementary Movie 2). In this study, we performed experiments in parallel in *C. elegans* and in mammalian cells. Where appropriate, the panels contain an icon to indicate the experimental system. For simplicity and coherence, we will use the mammalian nomenclature for the FERARI complex throughout the manuscript (Fig. 1c).

### Rab5-positive structures contact SNX1 compartments via kiss-and-run

FERARI contains two members that interact with Rab GTPases, Rab11FIP5 and Rabenosyn-5, which binds Rab5[8]. To investigate the role of Rab5 on SNX1-sorting endosomes, we imaged GFP-Rab5 and SNX1-mCherry in our model system, the *C. elegans* intestine (Fig. 1d). To our surprise, Rab5 vesicles contacted the SNX1 structures by kiss-and-run, in a very similar manner to what we had observed for Rab11 vesicles

(Fig. 1d and Supplementary Movie 3). These events were likewise observed between EHD1- and Rab5-positive structures (Fig. 1g). Moreover, even the periodicity of 7 s was the same than the one observed for Rab11 vesicles previously (Supplementary Fig. 1a). The kiss-and-run of Rab5 vesicles with the SNX1-sorting compartment was dependent on FERARI as knockdown of FERARI members reduced the residence time of Rab5 endocytic vesicles on SNX1 structures (Fig. 1e, f; Supplementary Fig. 3a, and Supplementary Movie 3). FERARI-mediated kiss-and-run of Rab5 vesicles with SNX1-sorting compartments is conserved in mammalian cells (Fig. 1h, I; Supplementary Fig. 3b and Supplementary Movie 4). Moreover, the periodicity was 4 s, the same as we observed for the Rab11 kisses (compare Fig. 1b, i). Our data indicate that Rab5 vesicles can also under undergo kiss-and-run on SNX1-sorting compartments.

### Early endosomes can undergo kiss-and-run with sorting compartments

We next asked whether the Rab5 compartments that undergo kiss-and-run are early endosomes. One of the hallmarks of early endosomes is that they can fuse with each other[9,10]. Thus, we analyzed our data for events in which Rab5 entities would undergo homotypic fusion and then contact SNX1-positive structures in a kiss-and-run event. Indeed, we observed Rab5 compartments fusing with each other and then go on to contact SNX1 compartments and finally move away to fuse with another early endosome (Fig. 2a and Supplementary Movie 5). To corroborate these results, we examined the behavior of another early endosomal marker, the ESCRT-0 cargo sorter component Hrs (HGRS-1 in *C. elegans*). We first confirmed that Rab5 and Hrs would co-localize on endosomal structures (Supplementary Fig. 4a, b and Supplementary Movies 6 and 7). Hrs-positive vesicles also contacted SNX1-sorting compartments, and the duration of the kiss was again dependent on FERARI (Fig. 2b–d; Supplementary Fig 4f and Supplementary Movie 8).

We and others have observed previously that SNX1 can co-localize with Rab5 and Rab7[7,11–13]. Given the results above, we revisited the issue and checked whether the SNX1-sorting compartments would still have some early endosomal identity using PI3P as a readout. We only observed a partial overlap of the PI3P marker 2xFYVE-GFP with SNX1, similar to what we observed for Rab5 and Hrs in *C. elegans* and mammalian cells (Supplementary Fig. 4c–e and Supplementary Movie 9[7]). Yet, we often observed globular Rab5, Hrs or 2xFYVE-GFP-positive globular structures, juxtaposed or even connected to SNX1-positive structures, consistent with the notion that these are sorting/maturing endosomes (Supplementary Movie 9).

### Kiss-and-run at SNX1-sorting endosomes is a common feature

Given our data above, we were wondering whether this mechanism of kiss-and-run on endosomal membranes is even more widespread, and asked whether other RAB GTPases could interact with FERARI. To this end, we performed a yeast-two-hybrid (Y2H) assay with Rab10, which is

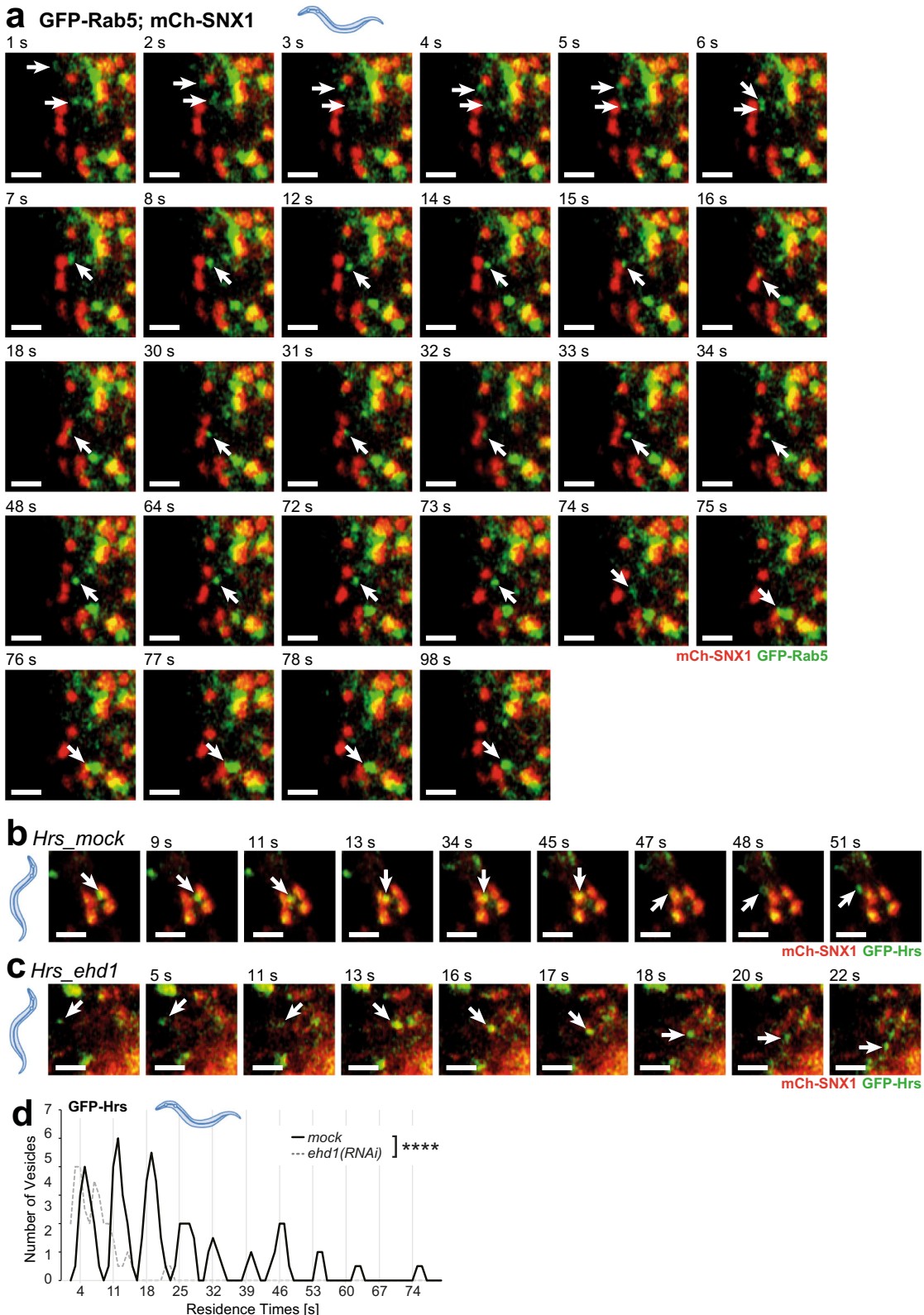

**Fig. 2 | Rab5 vesicles are fusing and doing kiss-and-run on the way to the sorting endosome, probably while carrying ESCRT-0 cargo adaptors. a** Movie stills for GFP-Rab5-positive vesicles (arrows) fusing, then docking on mCherry-SNX1 compartment (kiss-and-run for 18s) and finally fusing with a larger sorting endosome (with GFP-Rab5 and mCherry-SNX1 domains) (*n* = 53, kiss-and-run, *n* = 19 fusing). Scale bar: 2 μm. **b** Cargo adaptor ESCRT-0 subunit Hrs moves with vesicles that perform kiss-and-run on SNX1 compartments (*n* = 67) (movie stills from

Supplementary Movie 8, arrow points to Hrs-positive vesicle). Scale bar: 2 μm. **c** Kiss-and-run behavior of Hrs vesicles depends on FERARI subunit EHD1 (*n* = 30) (movie stills from Supplementary Movie 8, arrow points to Hrs-positive vesicle). Scale bar: 2 μm. **d** Hrs vesicles dock on SNX1 networks with residence times in distinct intervals. Groups of vesicles with similar residence times appear as peaks (see also Supplementary Fig. 4f for single-vesicle graph, *mock*: *n* = 67, *ehd1-KO*: *n* = 30). Unpaired two-tailed *t*-test: *P* = 2.03E-06.

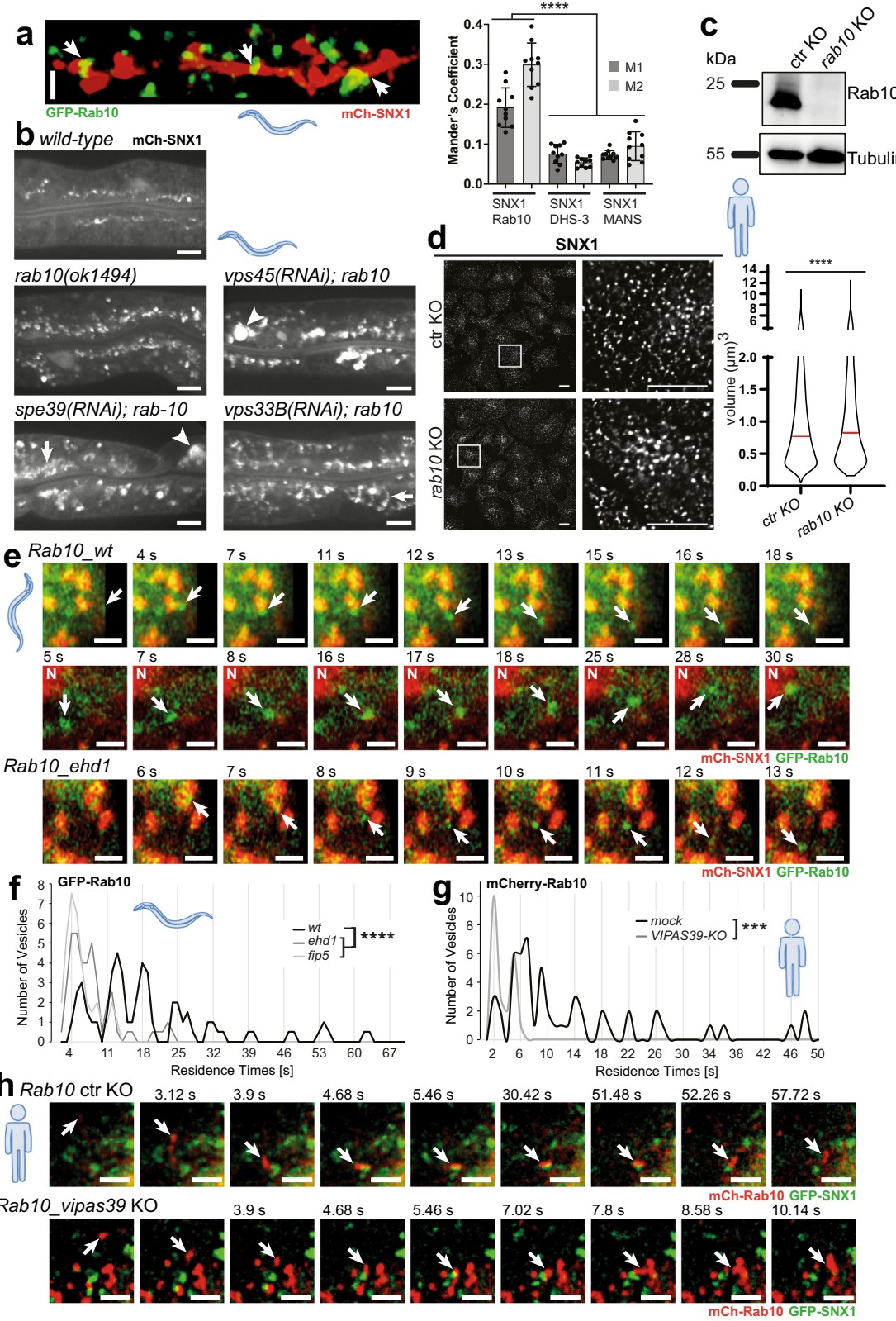

involved in recycling from sorting endosomes to basal-lateral membranes in polarized cells in *C. elegans* (Supplementary Fig. 5b)[14]. Rab10 interacted specifically with the FERARI member Rabenosyn-5. To further explore the possibility that Rab10-positive structures undergo FERARI-mediated kiss-and-run at sorting compartments, we first determined the localization of Rab10 with respect to SNX1. We observed Rab10 vesicular structures docked onto the SNX1

compartment (Fig. 3a and Supplementary Movie 10), similar to what we had observed for Rab11 and Rab5 previously[7]. Next, we explored the effect of loss of Rab10 on SNX1 (Fig. 3b and Supplementary Fig. 5a). Worms carrying the *rab10* loss-of-function allele *(ok1494)* showed an accumulation of SNX1 compartments that were drastically enlarged by knockdowns of *VPS45* and *VIPAS39*. VIPAS39 can also form a complex with VPS33B, which is called CHEVI[2]. To test whether the SNX1

**Fig. 3 | Rab10 vesicles undergo kiss-and-run with SNX1 compartments. a** GFP-Rab10 compartments docking onto mCherry-SNX1 networks in worm intestine (see also 3D projection, Supplementary Movie 10). Regions of co-localization are indicated by arrows. Quantification of co-localization by Mander's coefficients (n = 10). Data are presented as mean values +/− SD. M1 denotes the overlap between SNX1 and Rab10, and M2 between Rab10 and SNX1. One-way multiple comparison ANOVA test P-values: M1 SNX1-Rab10 vs. SNX1-DHS-3 P = 1.94E-08, M1 SNX1-Rab10 vs. SNX1-MANS P = 9.62E-09, M2 SNX1-Rab10 vs. SNX1-DHS-3 P = 4.37E-13, M2 SNX1-Rab10 vs. SNX1-MANS P = 4.43E-13. Scale bar: 2 μm. **b** Genetic interaction between FERARI subunits and Rab10. *rab10(ok1494)* causes accumulation of SNX1 compartments. RNAi of *vps45* and *vipas39* but not *vps33B* (CHEVI) show additional enlargement of SNX1 (arrowheads). n = 20 worms from 3 experiments (for quantification see Supplementary Fig. 5a). Scale bar: 10 μm. **c** Western blot of CRISPR−Cas9-mediated KO of Rab10 in HeLa cells (n = 3 independent experiments). **d** Size of the endogenous SNX1 structure is enlarged in *rab10* KO cells. SNX1 was detected by immunofluorescence. Violin plot showing the enlarged volume of the SNX1 structure in ctr and *rab10* KO cells (volume of >14,700 particles was measured from each group from three independent experiments, median indicated by the red line). Unpaired two-tailed *t*-test: P = 1.46E-17. Scale bars 10 μm. **e** Movie stills for Rab10 vesicles (arrow) showing kiss-and-run in *ehd1(RNAi)* (n = 40) and *wild-type* worms (n = 51) (Supplementary Movie 11). "N": nucleus with high mCh-SNX1 signal. Scale bar: 2 μm. **f** Rab10 vesicles show residence times with quantal increases (n = 51). This behavior is abolished in *ehd1* (n = 40) and *fip5* (n = 36) knock-downs (Supplementary Fig. 3a for single-vesicle plots). One-way multiple comparison ANOVA test P-values: *mock-ehd1* P = 7.23E-08, *mock-fip5* P = 4.29E-10. **g** Kiss-and-run of Rab10 on sorting compartments is conserved in metazoans. Residence times of Rab10 vesicles exhibit characteristic intervals abolished in *vipas39*-KO cells (see Supplementary Fig. 5c for single-vesicle plots). Unpaired two-tailed *t*-test: P = 0.000128. **h** Movie stills of mCherry-tagged Rab10 vesicles (arrow) showing kiss-and-run on SNX1 (GFP-tagged) compartment in ctr KO and *vipas39*-KO HeLa cells (n = 3 independent experiments, *mock*: n = 53, *vipas39-KO*: n = 22 vesicles) (see Supplementary Movie 12). Scale bar: 2 μm.

phenotype was specific for FERARI, we knocked down the CHEVI-specific component VPS33B in *rab10(ok1494)*. We did not observe any drastically enlarged SNX1 structures in *rab10(ok1494); vps33B(RNAi)* worms, implying that the effect was not due to a lack of CHEVI. Knocking out Rab10 in mammalian cells (Fig. 3c) resulted likewise in enlarged SNX1 structures (Fig. 3d). Moreover, Rab10 vesicles underwent kiss-and-run like Rab5 and Rab11 with similar periodicity, and this kiss-and-run was dependent on FERARI in *C. elegans* (Fig. 3e, f and Supplementary Movie 11) and mammalian cells (Fig. 3g, h; Supplementary Fig. 5c and Supplementary Movie 12).

Yet, not all endosomal RAB GTPases promote FERARI-dependent kiss-and-run on sorting endosomes. Unlike Rab5, Rab10 and Rab11, Rab7 did not interact with FERARI components in a Y2H assay (Supplementary Fig. 5b). Moreover, we never observed any kiss-and-run of Rab7 vesicles with the SNX1 compartment. Larger Rab7 compartments were stably connected to SNX1 and did not move around like the smaller vesicles, consistent with the notion that these structures are sorting/maturing endosomes (Supplementary Fig. 5d and Supplementary Movie 13). In contrast, homotypic fusion events between Rab7 vesicles were frequently observed (Supplementary Fig. 5e and Supplementary Movie 13). Similarly, we failed to observe kiss-and-run between Rab7 and SNX1 in mammalian cells (Supplementary Fig. 5f and Supplementary Movie 13). Taken together, our data provide strong evidence that three endosomal RABs - Rab5, Rab10, and Rab11-together with FERARI promote kiss-and-run of endosomes at SNX1-sorting endosomes in *C. elegans* and mammalian cells.

## FERARI interacts with distinct sets of SNAREs for Rab11 and Rab10-mediated recycling

Rab11 and Rab10 promote recycling to the apical and basal-lateral plasma membrane, respectively, in *C. elegans*[14,15]. Given that the apical and basal-lateral recycling pathways transport distinct cargoes, we would expect that Rab11 and Rab10 endosomes would use different SNAREs for fusion with sorting endosomes to provide a level of specificity for cargo recycling. We have shown previously that the syntaxins SYX6 and SYX7 are involved in the fusion of Rab11 vesicles with the SNX1 compartment[7]. Rab10 appears to use a largely non-overlapping set of SNAREs. The three different SNAREs *syx16, vamp7* and *vti1* indeed showed elongated Rab10 tubules characteristic of FERARI knock-downs, but had no effect on Rab11 compartments (Fig. 4a, b)[7]. In contrast, knockdown of the two SNAREs *syx6* and *syx7* had no effect on Rab10 endosomes. However, knockdown of the sorting endosomal SNARE SYX3 affected both Rab10 and Rab11 endosomes (Fig. 4a, c). Moreover, SYX3 partially co-localized with SNX1 tubular networks, Rab10 and the FERARI member EHD1 (Fig. 4d, e and Supplementary Movie 14). Collectively, our data suggest that FERARI uses one common SNARE and a selective set of SNAREs specific for Rab10 and Rab11 vesicles to promote fusion with the SNX1 compartment.

## Evidence for cargo flow from Rab5 endosomes through SNX1-sorting endosomes into Rab11 recycling endosomes

We surmise that during the kiss-and-run with SNX1-sorting compartment, cargo transfer would occur. In a first approximation we predicted that endosomal vesicles would either shrink (for Rab5) or increase (Rab11) and hence the fluorescence intensity of the endosomes would change. Thus, we measured the fluorescence intensity of Rab5 and Rab11-positive vesicles for their duration on sorting endosomes. We had observed previously that Rab11 vesicles before docking were less fluorescent than after leaving the SNX1 compartment[7]. Most Rab5 endosome lost fluorescence during the kiss, while we noticed the opposite for Rab11 endosomes (Fig. 5a, b and Supplementary Fig. 6c). Next, we repeated the analysis with human transferrin receptor (hTfR) and Glut1. We observed vesicles that increased in fluorescence intensity and vesicles that lost fluorescence in both cases, suggesting cargo uptake and loss, respectively (Fig. 5c, d and Supplementary Fig. 6d). Moreover, *rab11(RNAi)* abolished the hTfR containing vesicle population in which the fluorescence intensity increased during the kiss, supporting the model of cargo flow into Rab11 vesicles during the kiss (Fig. 5c). These data are consistent with vesicle fusion and cargo flow in and out of vesicles. We speculate that the increase in cargo fluorescence would happen in Rab11 vesicles and a decrease in Rab5 vesicles.

Still, we aimed to show the cargo flow more directly and co-expressed mApple-Rab5, GFP-SNX1 and Halo-tagged TfR in mammalian cells. We detected transfer of TfR from Rab5 to SNX1-positive compartments (Fig. 5e and Supplementary Movie 15). Similarly, repeating this experimental setup with mCherry-Rab11 allowed us to visualize the transfer of TfR from SNX-1 compartments into Rab11 vesicles (Fig. 5f and Supplementary Movie 15). Taken together, our data provide evidence for cargo transfer during kiss-and-run at SNX1-sorting endosomes.

## Available cargo amount contributes to the duration of the kiss between Rab11 and SNX1 compartments

We next wanted to test whether the recycling cargo can play a role in the duration of the kiss. We had noticed previously that cargo vesicles containing hTfR-GFP or Glut1-GFP showed extended residence times on SNX1 compartments compared to Rab11 vesicles, even though theses cargoes would leave the SNX1 compartment in Rab11 vesicles[7]. Therefore, we reduced cargo availability by downregulating hTfR-GFP and Glut1-GFP levels using weak RNAi against GFP (Supplementary Fig. 7a−d). The residence time of hTfR-GFP and Glut1-GFP vesicles was reduced and was comparable to the residence times that we had observed previously for Rab11, but the 7 s periodicity was not

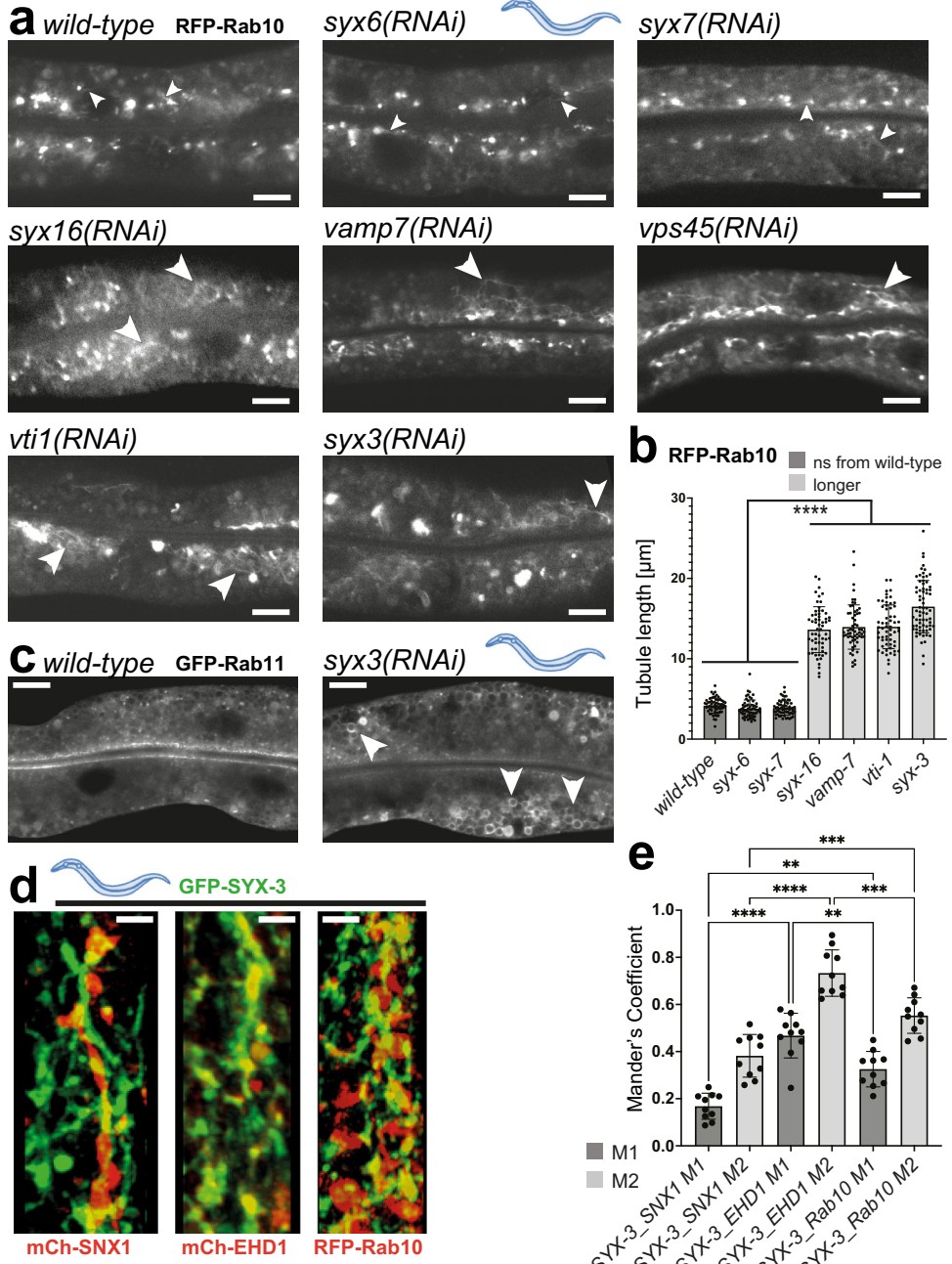

**Fig. 4 | Specific SNAREs are used to dock Rab10 vesicles through FERARI.**
**a** Knock-downs of *syx16, vamp7, vti1*, and *syx3* cause FERARI-like phenotypes as seen in *vps45(RNAi)* (long RFP-Rab10 tubules indicated by large arrowheads), while *syx6* and *syx7* do not (short tubules indicated by small arrowheads) (*n* = 20 animals per condition). **b** Tubule length was quantified for *n* = 10 different tubules in *n* = 6 animals examined over three independent RNAi experiments (total *n* = 60). Data are presented as mean values +/− SD. One-way multiple comparison ANOVA test *P*-values: all indicated differences are smaller than *P* = 1E-15. Scale bar: 10 μm. **c** Knock-down of *syx3* causes enlargement of Rab11 compartments, similar to previously

published FERARI knock-downs[7] (*n* = 20 animals per condition). Scale bar: 10 μm.
**d** GFP-SYX3 localizes to mCherry-SNX1, mCherry-EHD1, and RFP-Rab10 structures (Supplementary Movie 14 with 3D projections) (*n* = 10 animals). Scale bar: 2 μm.
**e** Mander's coefficients are shown for *n* = 10 worms. Data are presented as mean values +/− SD. One-way multiple comparison ANOVA test *P*-values: M1 SYX3-SNX1 vs. SYX3-EHD1 *P* = 1.0901E-09, M1 SYX3-SNX1 vs. SYX3-Rab10 *P* = 0.00116, M1 SYX3-EHD1 vs. SYX3-Rab10 *P* = 0.00424, M2 SYX3-SNX1 vs. SYX3-EHD1 *P* = 7.43E-12, M2 SYX3-SNX1 vs. SYX3-Rab10 *P* = 0.000351, M2 SYX3-EHD1 vs. SYX3-Rab10 *P* = 0.000149.

perturbed by the reduced cargo levels (Fig. 5g, h and Supplementary Fig. 7e−i[7],). We take this as an indication that cargo availability is one of the determinants for the duration of the kiss.

## SNX6 is required for efficient kiss-and-run
We have provided evidence above that cargo flows from early through sorting into recycling compartments. Thus, a mechanism has to be in place that provides directionality for cargo flow. Cargo sorting in

sorting endosomes is supposed to be regulated by multiple cargo adaptors[6,16]. Mammalian cargo adaptors SNX5 and SNX6 interact with SNX1 in a complex called ESCPE-1[17]. Knockout of SNX5 + 6 caused a reduction of SNX1 levels (Supplementary Fig. 1c−e)[16]. Thus, SNX5 and SNX6 are prime candidates for potential regulators of cargo sorting in the sorting endosome. *C. elegans* only expresses SNX6. Knock-down of SNX6 caused an increase of co-localization of Rab5, Rab11 and Rab10 with SNX1 (Fig. 6a−c and Supplementary Movies 16−18), probably due

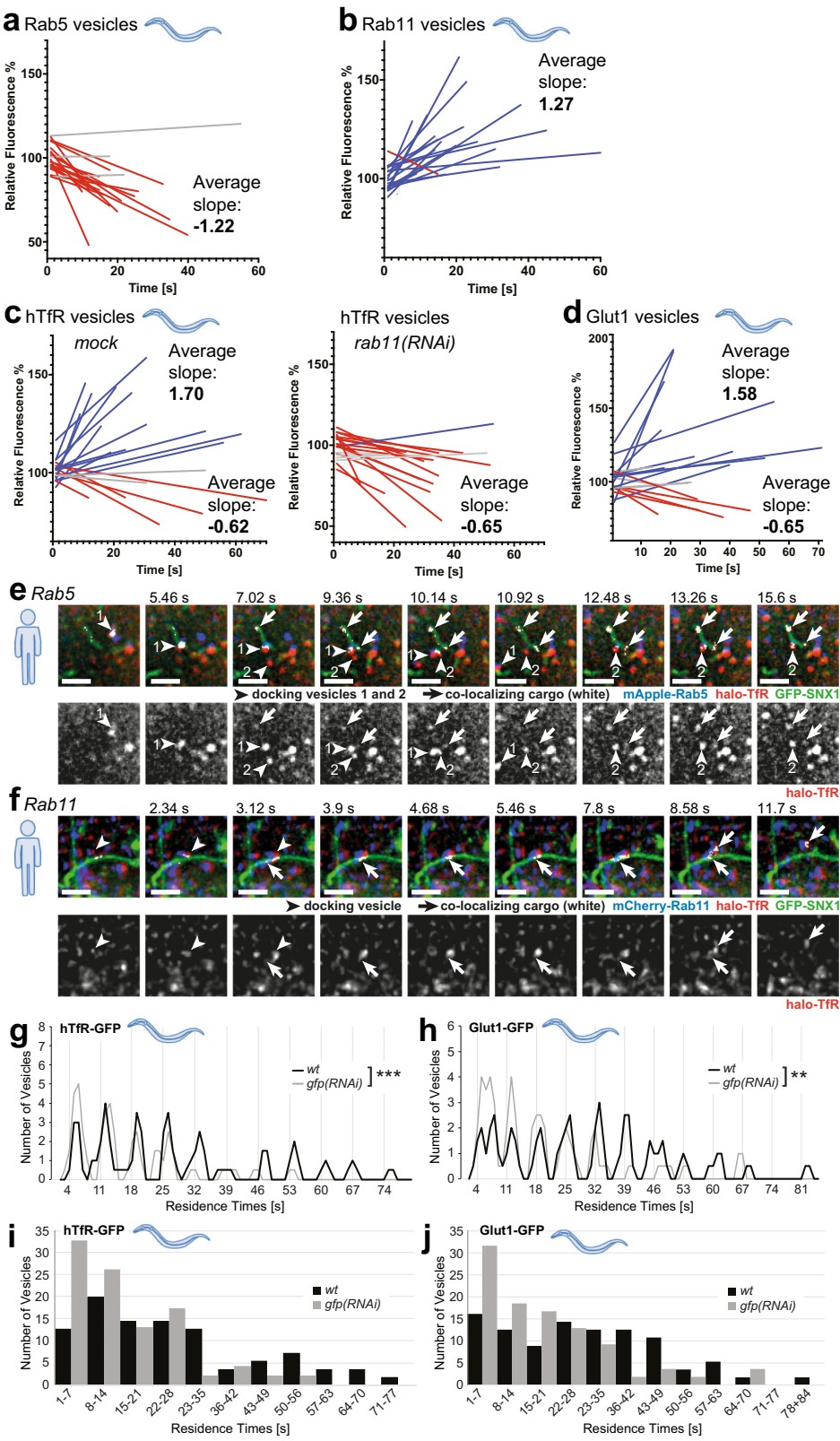

to an increase in residence time (Fig. 6d and Supplementary Fig. 3c). Likewise, an increase of co-localization between Rab10 and EHD1 was also observed, suggesting an involvement of FERARI (Fig. 6h). Consistent with a defect in cargo flow, Rab5 vesicles became enlarged and Rab11 vesicles smaller (Fig. 6e, f and Supplementary Fig. 3d, e). In addition, the fluorescence intensities of Rab5 and Rab11 vesicles did no longer change upon SNX6 knockdown (Supplementary Fig. 6a, b),

indicating the loss of cargo flow. Similarly, in *snx5+6* double knockout mammalian cells (Supplementary Fig. 1c), we observed an increase in co-localization between Rab5 and SNX1 (Fig. 6g and Supplementary Fig. 1f) and an increased volume of Rab5 and SNX1 structures (Fig. 6g). However, the co-localization of Rab11 with SNX1 was reduced and Rab11 vesicles became enlarged in *snx5+6* knockout cells (Fig. 7a, b, and c). This difference is likely due to the increased complexity in

**Fig. 5 | Cargo transfer during kiss-and-run events in worms and HeLa cells.**
**a** Fluorescence intensities of single vesicles were measured during the docking phase of kiss-and-run events. Linear regressions were calculated for Rab5 vesicles ($n = 20$), normalized and plotted together (single-vesicle graphs are shown in Supplementary Fig. 6c). **b** Rab11 vesicles were analyzed as described in **a**. **c** Cargo containing vesicles (hTfR-GFP) displayed either increasing or decreasing fluorescence intensities (see also Supplementary Fig. 6d for single-vesicle plots). **d** Analysis of Glut1-GFP-positive vesicles showed a similar result as for hTfR-GFP in **c**. **e** Cargo transfer from Rab5 vesicles into SNX1 compartments. Movie stills (from Supplementary Movie 15) showing halo-TfR (red) in docking Rab5 vesicles (blue) that is moving into a SNX1 tubule (green). Co-localizing pixels (shown in white and with arrows) were determined with the "co-localization finder" plugin in ImageJ. The halo-TfR channel is also shown for comparison. Two vesicles serially dock at the same site on the SNX1 tubule (shown by arrowheads 1 and 2). ($n = 3$ experiments with $n = 20$ cells each). Scale bar: 2 μm. **f** Rab11 vesicles can take up halo-TfR cargo

from SNX1 tubules. Rab11 vesicle (blue, arrowhead) docking on to a SNX1 tubule (green) taking up TfR (red). Co-localizing pixels (arrows pointing at white pixels) were determined by the "co-localization finder" plugin of ImageJ. Single-channel images for halo-Tfr are also shown for comparison. Stills are from Supplementary Movie 15. ($n = 3$ experiments with $n = 20$ cells each). Scale bar: 2 μm. **g** Residence times for hTfR-GFP vesicles decrease when cargo amount is reduced. Moving average for residence times is shown for *wild-type* (no knock-down, $n = 55$) and *gfp(RNAi)* ($n = 46$) (see Supplementary Fig. 7f, g for alternative graphs). Graphs with individual vesicle data are shown in Supplementary Fig. 7e. Unpaired two-tailed $t$-test: $P = 0.000655$. **h** Residence times for Glut1-GFP vesicles decrease when cargo amount is reduced. Residence times are plotted as in **a** for *wild-type* (no RNAi, $n = 56$) and *gfp(RNAi)* ($n = 54$). (see Supplementary Fig. 7h, i for alternative graphs). Unpaired two-tailed $t$-test: $P = 0.00268$. **i, j** Bar graphs with binning for each peak in **g** and **h**, showing the number of vesicles in each peak.

mammalian cells, as SNX5 and SNX6 can form also a different ESCPE-1 complex with SNX2[17]. For example, loss of SNX5/6 in RPE cells impaired CI-MPR retrograde transport to the TGN[18,19]. This role in retrograde transport is independent of the one that involves FERARI, as FERARI KO cells did not impede CI-MPR retrograde transport to the TGN (Fig. 7d, e). We can also not exclude that part of the difference we observed between *C. elegans* and mammalian cells might be due to different expression levels of the tagged SNX1. Despite these differences, our data imply a role of SNX5/6 in the coordination of cargo flow. Since cargo was probably routed with the help of SNX5/6 into recycling vesicles, we wondered whether SNX5/6 would interact with FERARI. SNX6, but not SNX5, bound to FERARI (Fig. 7f), and this interaction might be mediated by Rabenosyn-5 (Fig. 7g and Table 1). Taken together, our data suggest that SNX6 and FERARI cooperate in cargo flow in and out of sorting endosomes.

### Cargo flux at sorting endosomes depends on cargo adaptors

Our model of cargo flux would predict that cargo leaving the sorting endosome would need to be retained in recycling vesicles. Specific sorting nexins (SNX17 and SNX27) as well as AP1 have been described to promote recycling of cargo[20,21]. Consistently, we observed partial co-localization of AP1 with Rab11 but not with Rab5 (Supplementary Fig. 8a). Moreover, AP1 vesicles underwent kiss-and-run at SNX1-sorting endosomes (Fig. 8a, b; Supplementary Fig. 7j and Supplementary Movie 19). Thus, AP1 might be involved in the retention of cargo in Rab11 vesicles. To test this possibility, we knocked down the AP1 subunits APM1 and APS1 in *C. elegans*. As expected, SNX1 compartments were enlarged, with hTfR and Glut1 trapped in them (Fig. 8c, d and Supplementary Fig. 8b–d). Similarly, knockdown of SNX17 led to accumulation of hTfR in enlarged SNX1-positive compartments (Fig. 8c, black arrows and d, and Supplementary Fig. 8b, c). Moreover, we observed structures that were filled with SNX1, which also often contained hTfR (Fig. 8c, white arrowheads and asterisks). *snx27(RNAi)* did not lead to hTfR accumulations, indicating that it may not be involved in hTfR recycling to the plasma membrane. However, since we observed the enlargement of the SNX1-sorting compartment, we tested whether other cargoes were trapped under these conditions. Indeed, we found that Glut1 was trapped in SNX1 compartments, when SNX27 was missing (Supplementary Fig. 8c, d), consistent with data from mammalian cells[22,23]. Taken together, our data support a model in which cargo adaptors on recycling vesicles provide directionality of cargo flow during kiss-and-run.

### Discussion

In this study, we show that the tethering complex FERARI mediates kiss-and-run of at least three types of vesicles/endosomes with the SNX1-sorting compartment in *C. elegans* and mammalian cells. During these encounters, cargo is transferred from one compartment to the

next. We speculate that the amount of transferable cargo may contribute to the duration of the kiss.

We envisage a model (see Fig. 8e) in which Rab5 endocytic vesicles/early endosomes on their way from the plasma membrane inwards kiss SNX1-sorting endosomes. During this kiss, cargo destined for immediate recycling could already exit the Rab5 compartment. We predict negative selection in this process in that ubiquitinated cargo, which interacts with the ESCRT-0 component Hrs would have to remain in the Rab5 compartment, while un- or deubiquitinated cargo would be free to leave. Possibly, attraction by the negative membrane curvature during the kiss facilitates the movement of membrane proteins and specific lipids towards the sorting endosome[24,25]. Backflow of cargo would be prevented by its binding to the cargo adaptor SNX6 and potentially other cargo adaptors in the sorting compartment. In fact, we found that loss of function of SNX17 and SNX27 resulted in enlarged SNX1 compartments and cargo accumulation. SNX17 and SNX27 were shown to be involved in recycling through retriever and retromer recycling pathways[20,21]. Combinatorial low affinity interactions of cargo with different adaptors may drive sorting in the SNX1-sorting compartment[26]. The efflux of cargo would occur into Rab10 or Rab11 vesicles, that would kiss in a similar way than Rab5 endosomes. In the case of Rab11, we surmise that AP1 is critical to retain the cargo in the Rab11 compartment and avoid backflow into the sorting compartment. The presence of AP1 on tubular recycling endosomes has been shown previously[4]. The retention could be helped by either the concentration of AP1 or a higher affinity of the cargo for AP1 than for a SNX cargo adaptor.

Since the binding affinities during the sorting process are relatively low, mis-sorting is prone to happen. Kiss-and-run of Rab5 endosomes on their journey inward and of Rab11 and Rab10 endosomes outward provide opportunities to correct mistakes in sorting and may increase fidelity of the entire process. Moreover, sorting is not perfect in that there is always some bulk flow. This phenomenon is probably best recognized at the endoplasmic reticulum, where ER-resident proteins escape routinely in COPII vesicles to the Golgi. The cell has even evolved a retrieval system to counteract this loss of ER proteins from the ER[27–29]. Thus, kiss-and-run would make the endocytic system more robust and less error-prone. We often observed that Rab5 and Rab11 endosomes would contact several SNX1 endosomes for various times and presumably engage in cargo exchange. We speculate that during these repeated contacts more and more cargo would be transferred from one compartment to the next until the Rab5 vesicles may have already transferred the bulk of cargo destined to the immediate return to the plasma membrane. Rab11 vesicles could take up more cargo and at the same time might have backflow of cargo that was missorted into the Rab11 vesicle. Thus, these kiss-and-run events may contribute to the fidelity of cargo sorting in the endosomal system, similar to a distillation process. This model is not without precedence as iterative

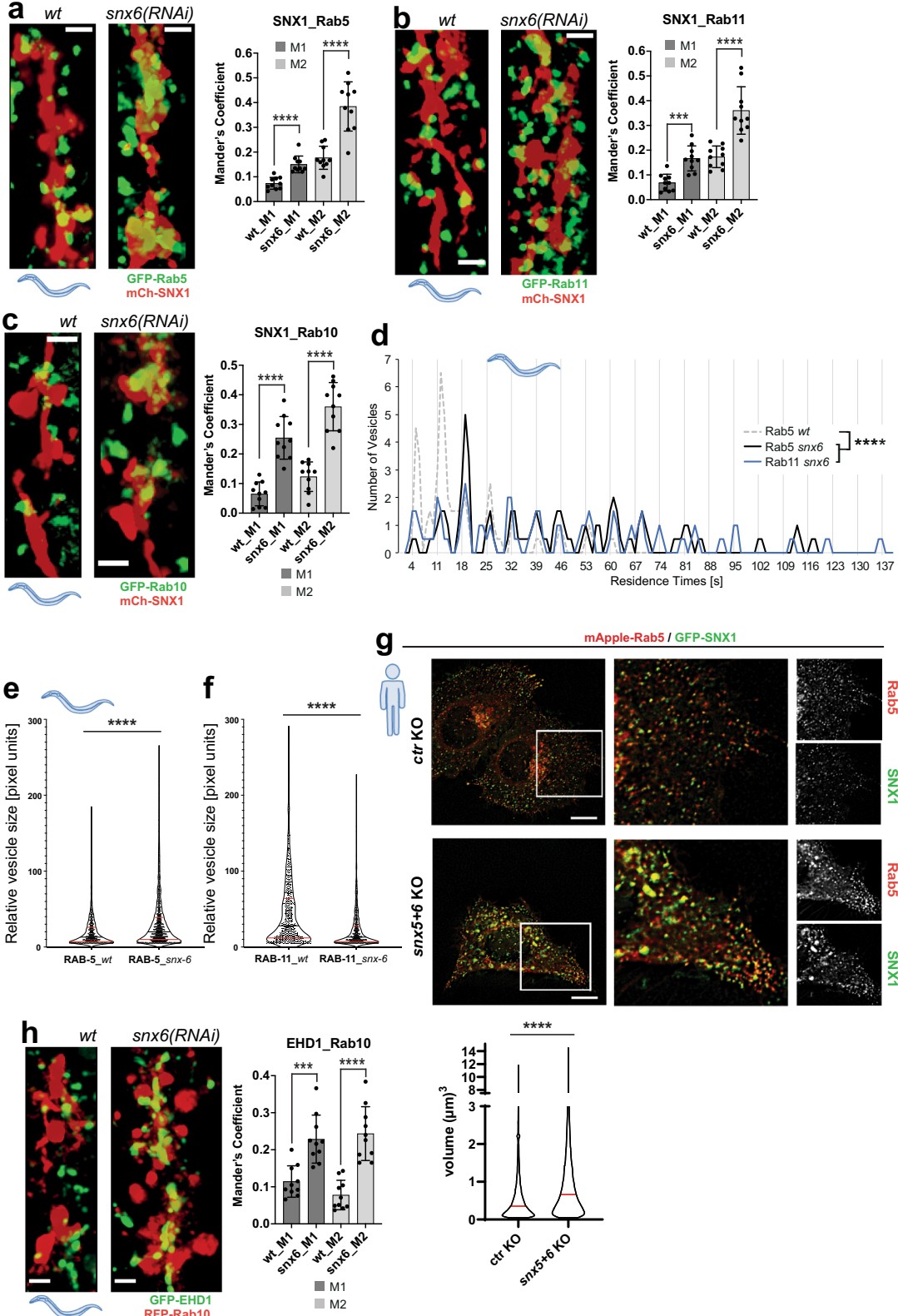

fractionation of recycling receptors in sorting endosomes was already proposed in the late 1980s[30], which was based on the proposed distillation model for transport through the Golgi apparatus in which newly synthesized proteins would be sorted away from Golgi resident proteins during transport through the Golgi[31]. The distillation model was later modified to explain Golgi maturation, which is akin to endosome maturation.

Our data extend the current view on how recycling from early endosomes to the plasma membrane is achieved. The prevailing model posits that SNX1 forms tubules on Rab5-positive early endosomes and that these can become Rab-positive and subsequently pinch off to go to the plasma membrane or the TGN. While we also observe such events, kiss-and-run of Rab5- or Rab11-positive structures on SNX1 compartments also occurred. Whether the SNX1 compartment is

**Fig. 6 | Effects of cargo adaptor SNX6 on Rab5, Rab11, and Rab10 vesicles in worms and HeLa cells. a** *snx6(RNAi)* leads to increased co-localization of GFP-Rab5 vesicles with mCherry-SNX1 networks (see Supplementary Movie 16). Mander's coefficients ($n = 10$). Data are presented as mean values +/− SD. Unpaired two-tailed *t*-test: M1 $P = 2.61E$-05, M2 $P = 2.06E$-05. Scale bar: 2 μm. **b** *snx6(RNAi)* leads to increased co-localization of GFP-Rab11 vesicles with mCherry-SNX1 networks (see Supplementary Movie 17). Mander's coefficients ($n = 10$). Data are presented as mean values +/− SD. Unpaired two-tailed *t*-test: M1 $P = 0.000113$, M2 $P = 9.501E$-05. Scale bar: 2 μm. **c** *snx-6(RNAi)* increases GFP-Rab10 compartment co-localization with mCh-SNX1 in *C. elegans* (Supplementary Movie 18). Mander's coefficients ($n = 10$). Data are presented as mean values +/− SD. Unpaired two-tailed *t*-test: M1 $P = 3.96E$-06, M2 $P = 1.16E$-06. Scale bar: 2 μm. **d** *snx6(RNAi)* drastically increases the residence times of vesicles. The Rab5 *wild-type* vesicles are plotted as a dashed line for comparison (see Supplementary Fig. 3c for single-vesicle plots). Please note the elongated *x*-axis to accommodate vesicles with very long residence times ($n = 62$

for Rab5, $n = 57$ for Rab11). One-way multiple comparison ANOVA test *P*-values: *Rab5 wt* vs. *Rab5 snx6* $P = 7.76E$-07, *Rab5 wt* vs.*Rab11 snx6* $P = 1.051E$-06. **e** Rab5 vesicles from *snx6(RNAi)* worms are increased in size (*wt*: $n = 828$, *snx-6*: $n = 1459$) (see Supplementary Fig. 3d for single worms). Unpaired two-tailed *t*-test $P = 1.45E$-11. **f** Rab11 vesicles from *snx6(RNAi)* worms appear smaller (*wt*: $n = 774$, *snx6*: $n = 995$) (see Supplementary Fig. 3e for single worms). Unpaired two-tailed *t*-test $P = 1E$-15. **g** Co-localization of Rab5 and SNX1 is increased in *snx5 + 6* KO cells. HeLa cells were co-transfected with GFP-SNX1 and mApple-Rab5. Representative images from live-cell imaging of ctr and *snx5 + 6* KO are shown. Rab5 and SNX1-positive structures are enlarged in *snx5 + 6* KO cells ($n = 3$ independent experiments, see quantification of volume of Rab5 structures below). Unpaired two-tailed *t*-test $P = 1.615E$-249. Scale bars 10 μm. **h** Knock-down of *snx-6* increases RFP-Rab10 compartment co-localization with GFP-EHD1 in *C. elegans*. Mander's coefficients ($n = 10$). Data are presented as mean values +/− SD. Unpaired two-tailed *t*-test: M1 $P = 0.000275$, M2 $P = 1.896E$-05. Scale bar: 2 μm.

always part of an endosome that matures from early-to-late[11,13] or whether it represents a separate sorting compartment (i.e., endosomal recycling compartments)[3,32,33] that was formed as a result of a pinching process from a maturing endosome remains an open question. Moreover, the equilibrium between kiss-and-run and Rab11-tube pinching might be tissue and cell type-dependent or dependent on the endocytic and secretory capacity and demand at any given point in time.

We show that cargo can contribute to the length of the kiss of the Rab11 endosomes with sorting endosomes. One possibility for this contribution could be that cargo could get stuck in the fusion stalk between the two compartments and interfere with the fission process. However, further studies are needed to test this model.

Taken together, we provide strong evidence for an endocytic cargo sorting mechanism through FERARI-mediated kiss-and-run, which together with the previously established recycling system provides robustness and fidelity to the system. The cargo flow determined by cargo adaptors and kiss-and-run provides a simple and efficient mechanism that allows for cargo sorting mistakes and step-by-step enrichment of specific cargo into several cargo carriers. However, many open questions remain about the coordination of the different fusion and fission processes by FERARI, in terms of specificity and also temporal and spatial control, which will be subject of future studies.

## Methods
### Worm husbandry
*C. elegans* worms were grown and crossed according to standard methods[34]. RNAi was performed as previously described[35]. All experiments were carried out at 20 °C, and worms were imaged at the young adult stage (with only few eggs).

The following worm strains and transgenes were used in this study: *pwIs206[vha6p::GFP::rab-10 + Cb unc-119(+)], pwIs782[Pvha-6::mCherry::SNX-1], pwIs414[Pvha-6::RFP::rab-10, Cb unc-119(+)], dkIs218[Popt-2-GFP-syx-3; Cb unc-119(+)], pwIs621[vha-6::mCherry-RME-1], pwIs72[vha6p::GFP::rab-5 + unc-119(+)], pwIs170[vha6p::GFP::rab-7 + Cb unc-119(+)], pwIs90[Pvha-6::hTfR-GFP; Cbr-unc-119(+)], qxEx2247 [Pvha-6::Glut1::GFP], pwIs69[vha6p::GFP::rab-11 + unc-119(+)], pwIs87 [Pvha-6::GFP::rme-1; Cbr-unc-119(+)], [Pdhs-3::dhs-3::GFP], pwIs481[Pvha-6::mans-GFP, Cbr-unc-119(+)], pwIs518[vha-6::GFP-HGRS-1], pwIs846 [Pvha-6-RFP-rab-5; Cb unc-119(+)], rab-10(ok1494), Is(pvha-6::PI3P:GFP), [GFP::2xFYVE].*

All "Is" markers denote stably integrated arrays, the exception is qxEx2247, which was not integrated but showed a very high transmission and expression throughout the intestine. All promoters showed good and exclusive expression in gut cells.

### Microscopy of *C. elegans*
Live microscopy on worms was performed as described[7,35]. In short, worms were immobilized on 2% agarose pads on microscopy slides

using levamisole (50 mM); coverslips were sealed using Vaseline. Overview images were acquired with an Olympus Fluoview FV3000 system using a high-sensitivity spectral detector (HSD) at a standard voltage setting (PTM) of 500. For higher resolution, the Galvano scan device was applied. A 60x objective with silicone oil was used, resulting in a pixel size of 0.098 μm. Laser intensities were at 4–10% for both 488 nm (GFP) and 561 nm (RFP, mCherry) wavelengths. Sampling speed was 8.0 μs/pixel with a zoom factor of 2.1. All images for corresponding experiments were processed with the same settings to insure comparable results.

High-resolution 3D images and movies were obtained with a Zeiss LSM 880 microscope with Airyscan capabilities. The fast mode in the Zen Black software was used for all images. For movies, resolution was traded for speed by reducing the averaging to 2–4x, resulting in the required frame speeds of 0.5–1.0 s to follow vesicles. To catch high enough numbers of vesicles, a region of approximately 70 μm² was covered (about 2 intestinal cells). Movement could be observed up to 30–45 min after immobilization of worms. From these overview movies, smaller regions of 70 × 70 pixels were selected, showing only 1–2 vesicles and events. Since vesicles and tubules are very thin, they often appear only slightly above background. Brightness and contrast were adjusted to allow visibility of the faint vesicles and to avoid random background noise that will interfere with the visualization of the processes. This may cause the images to appear oversaturated, but the original movies were taken at low laser settings to avoid bleaching and so oversaturation was never a problem. These movies were then quantified. We observed persistent movement in many cells, and events were mainly limited by the use of only one imaging plane (due to speed limitation of the microscope). The "StackReg" plugin in Fiji was used to get rid of worm shaking and drifting motions. "Bleach correction" was used to avoid irritating blinking during repeated viewing of movies (overall, bleaching was minimal due to low laser settings).

Residence times were measured by scanning through single movies and counting the frames from initial tethering to the moment the vesicle left. In worms the frames were acquired at 1 frame per second and so frames and seconds correspond to 1:1, while in HeLa cells 1 frame corresponded to 0.78 s, resulting in non-integer numbers. Vesicles with the same residence times were binned and plotted as graphs, where groups with similar residence times appear as peaks.

### Compartment quantifications in worms
Rab10 tubule length was measured with the freehand line ROI function in Fiji. Since only one *z*-plane was used, this leads to an underestimation of the true length of tubules, but since FERARI phenotypes were exceptionally strong, no further measurements were deemed necessary. Mander's Coefficients for co-localization between compartments were measured with the JACoP plugin of

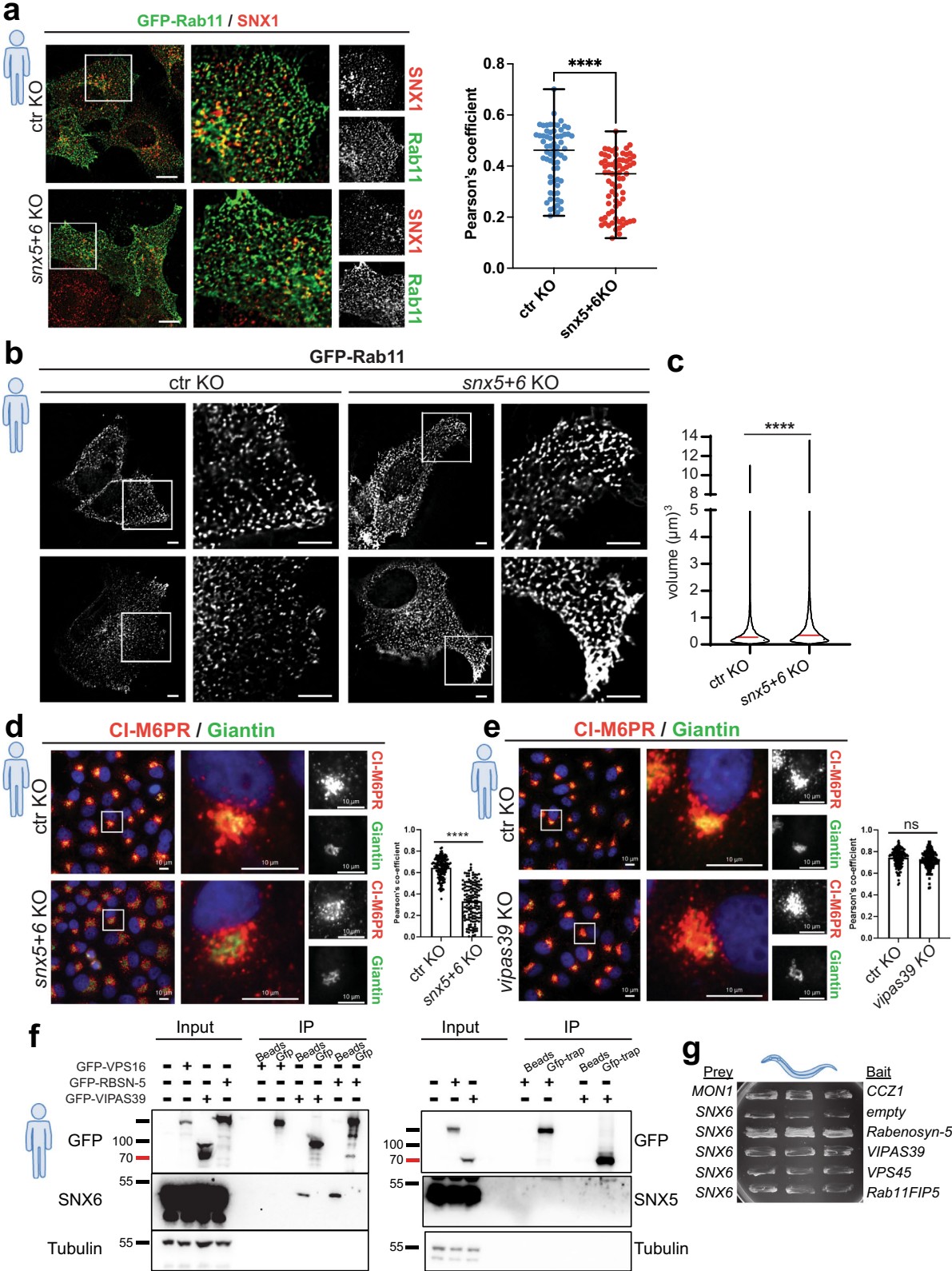

Fiji, using all available *z*-planes over whole intestinal cells for 10 worms each. M1 denotes the overlap of marker 1 with marker 2, while M2 measures the overlap of marker 2 with marker 1. Vesicle size for Rab5 and Rab11 was measured using "analyze particles" in Fiji in movie stills, because vesicle movement during the recording of *z*-stacks makes it impossible to measure their size.

**Cargo transfer measurements**

Stable vesicles were chosen during kiss-and-run events to measure overall vesicle fluorescence. A circular ROI was chosen to include the vesicle and the "mean gray area" was measured in each frame during the docking phase. The ROI was moved in each frame to contain the vesicle and to compensate for small movements during the movie.

**Fig. 7 | SNX6 cargo adaptor affects recycling in HeLa cells and interacts directly with FERARI. a** Co-localization between GFP-Rab11 and endogenous SNX1 is reduced in *snx5 + 6* double knock out cells. Antibodies against GFP and SNX1 were used to detect Rab11 and SNX1 by immunofluorescence, respectively. Scale bars 10 μm; 5x magnified. Pearson's coefficient from *n* = 65 and *n* = 71 cells from ctr KO and *snx5 + 6* KO cells, respectively. Data are presented as single data points with median and min to max whiskers. Unpaired two-tailed *t*-test *P* = 2.5316E-07. **b** Size of the GFP-Rab11-positive structures is increased in *snx5 + 6* KO cells in comparison with the ctr KO cells (*n* = 3 independent experiments) Scale bars 10 μm; magnification 5x. **c** Volume of the GFP-Rab11-positive particles in both ctr and KO cells were determined (volume of >15,000 particles were measured from each group from three independent experiments, median is shown by the red line). Unpaired two-tailed *t*-test *P* = 3.05E-73. **d** CI-MPR trafficking is impaired in *snx5 + 6* KO cells. CI-

MPR (red) is mostly localized in TGN (green) area in ctr KO cells and is dispersed in *snx5 + 6* KO cells (*n* = 3 independent experiments, *n* = 150 cells per condition). Pearson's coefficient measurements are shown on the right. Unpaired two-tailed *t*-test *P* = 1.97E-51. Scale bars 10 μm. **e** CI-MPR localization remain unchanged in *vipas39* KO cells (*n* = 3 independent experiments, *n* = 150 cells per condition). Scale bars 10 μm; magnification 5x. Co-localization was quantified by Pearsons's coefficients. Unpaired two-tailed *t*-test *P* = 0.0536. **f** Immunoprecipitation data showing that endogenous SNX6 (left panel), but not SNX5 (right panel), binds to FERARI members Rabenosyn-5 and VIPAS39 in HEK-293 cells (*n* = 3 independent experiments). **g** *C. elegans* SNX6 interacts with RABS-5 (but not with other FERARI subunits) in Y2H assays. Quantification is shown in Table 1 (*n* = 3 growth experiments with *n* = 6 independent yeast colonies each).

## Table 1 | Y2H (yeast-two-hybrid) interactions

| Prey | Bait | Growth |
|------|------|--------|
| *sand-1* | *ccz-1* | ++ |
| *snx-6* | *empty* | a |
| *snx-6* | *rabs-5* | +++ |
| *snx-6* | *spe-39* | a |
| *snx-6* | *vps-45* | a |
| *snx-6* | *rfip-2* | a |

ªSlight auto-activation.

Brightness plots of each vesicle were analyzed by linear regression. Data was normalized to allow comparison of the slopes and to determine a net change of fluorescence intensity over the time of vesicle docking. All linear regression lines were plotted together and average slopes were calculated using GraphPad Prism. A good control for photobleaching is the analysis of strains with *snx6(RNAi)*, where vesicles are docking over longer times but show very flat profiles (see Supplementary Fig. 6a, b).

Cargo transfer during kiss-and-run was visualized in HeLa cells by taking 3-color movies with halo-TfR (cargo), mApple-Rab5 or mCherry-Rab11 (vesicle) and GFP-SNX1 (tubular compartment). For this, stably expressing mApple-Rab5 HeLa cells were co-transfected with Halo Tfr and GFP-SNX1 plasmids for 48 h followed by treatment with 200 nM of Janelia Fluor® 646 HaloTag® Ligands for 10 min. Cells were washed twice with fresh media, followed by live-cell imaging with imaging buffer. In case of cargo trafficking from SNX1 to Rab11 vesicles, cells were triple-transfected with mCherry-Rab11, Halo-TfR and GFP-SNX1. In some cases, the cargo transfer from Rab5 vesicles into SNX1 (or from SNX1 into Rab11) was apparent. Since the complexity of 3-color movies makes it difficult to follow the cargo the "co-localization finder" plugin of Fiji was used to highlight the pixels with transferred cargo (only the relevant co-localization is shown for clarity).

### GFP-Knock-in at the Rab11 locus in HeLa cells

A GFP construct was inserted into the N-terminus of Rab11 at the endogenous level by using CRISPR/Cas9 homology directed repair. The following strategy was taken to insert GFP into Rab11 at the genomic level (Supplementary Fig. 9). Two guide RNAs were designed from introns before and after the first exon of Rab11. Annealed oligonucleotides were cloned into two different plasmids, Px458 mCherry (kindly provided by Mirjam Pennauer), and Px459 Puro (Addgene), respectively. As indicated in the scheme, five PCR products were synthesized. GFP was synthesized by using a GFP containing plasmid as a template and for the rest, genomic DNA from HeLa cells served as template. All these PCR products were cloned into pUC19 plasmid (Addgene; 50005, kind gift from Dr. Shahidul Alam, Pharmazentrum) by Gibson assembly. After transformation, sequencing confirmed the insertion of the template into the vector. Two gRNAs containing vectors and the template

containing vector were then transfected into HeLa cells by using the Helix-in (OZ biosciences) transfecting reagent according to the manufacturer's protocol. After seven days of transfection, cells were FACS sorted, and GFP+ cells were collected. For the confirmation of the GFP-knock-in, PCR and western blot were performed. For all experiments, heterozygous GFP-Rab11/Rab11 cell lines were used, unless cells were transiently transfected with mCherry-Rab11, where indicated.

### Cell culture, transfection, and CRISPR–Cas9 KO in mammalian cells

HEK293 and HeLa cells were cultured and maintained in DMEM (Sigma) high-glucose medium with 10% FCS (Bioconcept), penicillin–streptomycin (1%), sodium pyruvate and L-glutamine. Both cell lines have been authenticated by STR analysis by Microsynth AG (Balgach, Switzerland). Cells were plated 1 day before transfection at 60–70% confluency and later transfected for 48 h using the Helix-in (OZ biosciences) transfection reagent according to the manufacturer's instructions. One to 2 μg of DNA was used per reaction, based on a 10-cm dish. For CRISPR–Cas9-mediated KO, guide RNAs were selected using the CRISPR design tools (http://chopchop.cbu.uib.no/ and https://www.benchling.com/). A list of oligonucleotides is provided in Supplementary Table 1. Two guide RNAs were designed from two different exons for each target gene. Annealed oligonucleotides were cloned into Px458 GFP and Px459 Puro. In brief, HeLa cells were seeded at 2 x10⁶ cells per 10-cm dish. The following day, cells were transfected with 2.5 μg of the plasmids (control vectors without insert or vectors containing a guide RNA against the target gene). Transfecting medium was exchanged with fresh medium after 4 h. Cells were treated with puromycin for 24 h after transfection followed by FACS sorting (for GFP+ cells) the next day. For FACS sorting after 48 h of transfection, cells were trypsinized and resuspended in cell-sorting medium (2% FCS and 2.5 mM EDTA in PBS) and sorted on a BD FACS AriaIII Cell Sorter. GFP + cells were collected and seeded into a new well.

### Immunoprecipitation assays

HEK293 cells were co-transfected with the indicated DNA constructs. After 36–48 h of transfection, protein extracts were prepared in lysis buffer (50 mM Tris/HCl pH 7.5, 150 mM NaCl, 1% NP-40) containing Halt protease inhibitor cocktail (Thermo Scientific; 186 1279) at 4 °C for 20 min followed by centrifugation at 4 °C for 20 min at 18,000 x g. Immunoprecipitations were performed as previously described[7]. In brief, protein extracts were incubated with GFP-Trap_A beads (nanobodies for GFP; gta-20-chromotek), for 6 h at 4 °C with rotation, and then washed five times with lysis buffer (1 ml). Protein complexes were eluted by heating beads for 5 min at 95 °C in 2x sample buffer and resolved by SDS–PAGE on 10% and 12.5% gels followed by immunoblot analysis. Blots were developed using Amersham ECL Prime Western Blotting Detection Reagent (RPN2236) and X-ray film (Amersham Hyperfilm ECL-28906839).

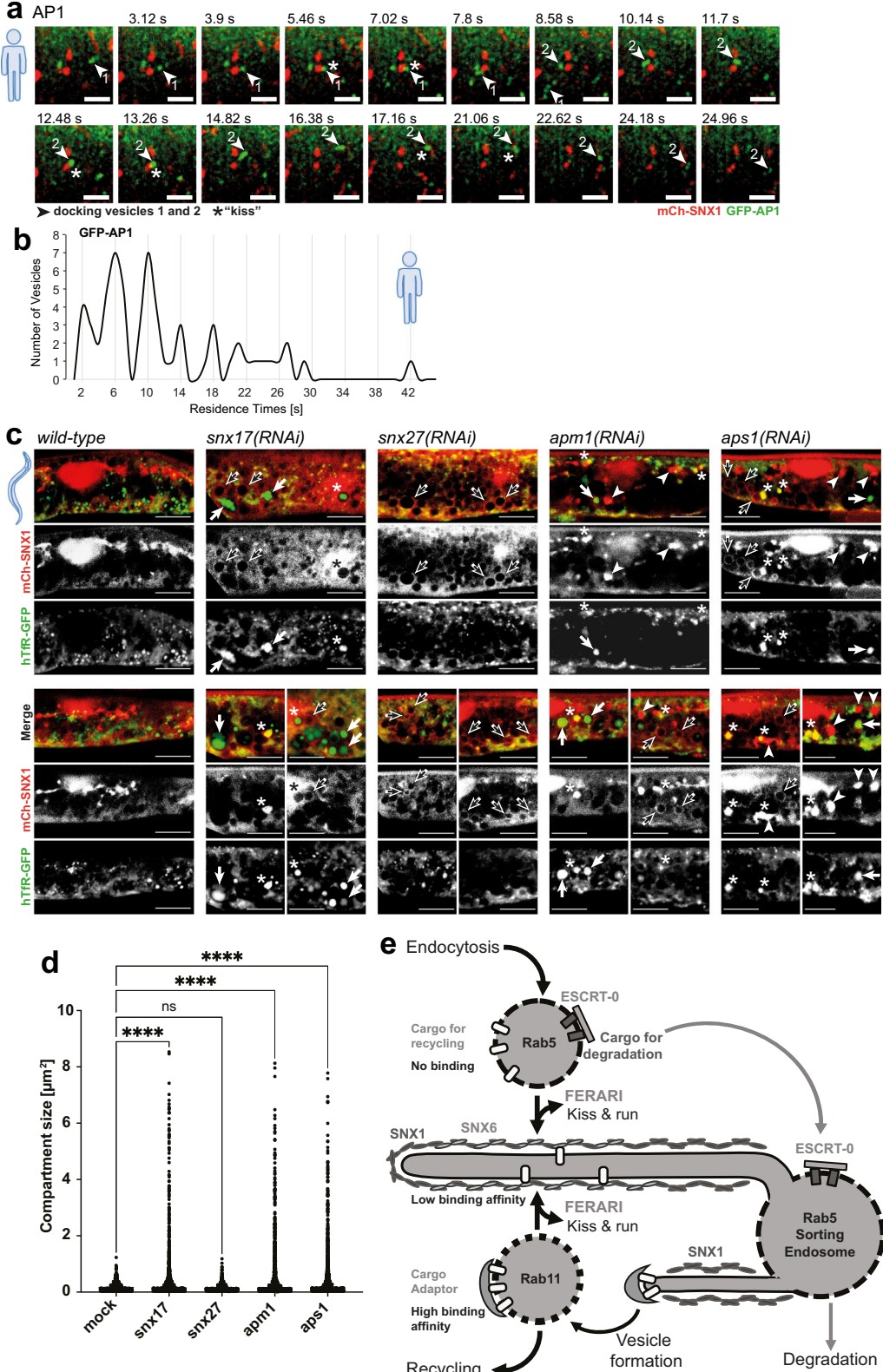

**a** AP1

➤ docking vesicles 1 and 2   ✱ "kiss"   mCh-SNX1 GFP-AP1

**b** GFP-AP1

**c** wild-type   snx17(RNAi)   snx27(RNAi)   apm1(RNAi)   aps1(RNAi)

mCh-SNX1   hTfR-GFP   Merge   mCh-SNX1   hTfR-GFP

**d**

**e** Endocytosis

## Western blot analysis

Cells were collected and lysed in lysis buffer (50 mM Tris/HCl pH 7.5, 150 mM NaCl, 1% NP-40) containing a protease inhibitor cocktail (Roche). Protein concentrations were determined in all experiments using the Bio-RAD protein assay (Bio-RAD, 500-0006) and 20–40 µg of total protein was loaded onto SDS–PAGE gels ranging between 10% and 15% acrylamid before transfer onto nitrocellulose membranes (Amersham Protran; 10600003). Membranes were blocked with 5% milk, 0.1% Tween20 for 1 h at room temperature. The primary antibody incubation was overnight at 4 °C and the secondary HRP-coupled antibodies were incubated for 1 h at room temperature. For RAB10, SNX5 and SNX6 detection, primary antibodies were diluted in Can Get Signal™ solution 1 and incubated with the blot overnight. The secondary HRP-coupled

**Fig. 8 | Effects of cargo adaptors on cargo flow. a** AP1 cargo adaptor moves with vesicles performing kiss-and-run on SNX1 compartments. Movie still (from Supplementary Movie 19) showing docking of vesicle 1 (arrowhead), followed by a second vesicle (arrowhead 2) that docks at the same site and then moves on to dock to an additional SNX1 tubule ($n = 61$). Docking of vesicles is indicated by asterisks. Scale bar: 2 µm. **b** AP1 vesicles show residence times with 4 s intervals (as seen with other vesicles docking in a FERARI-dependent manner). Residence times of $n = 61$ vesicles were measured and plotted in bins (see Supplementary Fig. 7j for single-vesicles graph). **c** Adaptor knock-downs affect cargo traffic through SNX1 compartments. RNAi of *snx17, snx27, apm1* or *aps1* cause enlargement of SNX1 compartments (black arrows: large empty round compartments, arrowheads: large filled compartments (see also Supplementary Fig. 8b)). hTfR-GFP was found in enlarged compartments as well (white arrows). Single-channel images are also shown. hTfR accumulations were absent in *snx27(RNAi)* worms. Some of the hTfR accumulations co-localized with enlarged SNX1 structures (asterisks). Experiments

were repeated $n = 3$ times with $n = 20$ worms analyzed. Scale bar: 10 µm. **d** Quantification of enlarged hTfR-GFP compartments in adaptor knock-down worms ($n = 6$). One-way multiple comparison ANOVA test $P$-values: all differences are smaller than $P = 1E\text{-}15$, ns $P = 0.967$. **e** Cargo flow hypothesis, showing 3 subsequent steps of cargo sorting. First, Rab5 vesicles unload non-bound recycling cargo and enrich ESCRT-bound cargo for transport into late endosomes/lysosomes. Second, low affinity binding to SNX6 organizes cargo inside the SNX1 tubular network. Third, higher affinity adaptors in Rab11 vesicles are used to bind cargo for final transport to destination (e.g., plasma membrane). Subsequent steps of kiss-and-run ensure enrichment of wanted cargoes (and loss of unwanted ones), allowing for a "proofreading" of cargo content despite low binding affinities to adaptors. The process of vesicle biogenesis by formation of tubules, followed by cargo enrichment with adaptor complexes and pinching-off of vesicles is a crucial first step in the sorting process and is also depicted.

---

antibodies were diluted in Can Get Signal™ solution 2, and the incubation was performed for 1 h. The blots were developed using the western Blotting detection kit WesternBright™ ECL (advansta; K-12045-D50) and the Fusion FX7 (Vilber Lourmat) image acquisition system.

### Immunostaining in mammalian cells

Cells were plated onto sterile 13-mm glass coverslips. Cells were fixed with 2% paraformaldehyde for 15 min, permeabilized (0.1% Triton X-100 in PBS) for 5 min and blocked with 2% BSA containing 5% goat serum in PBS for 1 h. Coverslips were incubated in primary antibodies for 2 h and washed five times in PBS followed by a 1-h incubation with fluorescently tagged secondary antibodies. After secondary antibody incubation, coverslips were washed further five times in PBS and mounted onto glass slides using Fluoromount-G (Southernbiotech; 0100-01). Images were taken with an inverted Olympus FV3000 confocal microscope (Olympus, Wallisellen, Switzerland) using a Plan Apochromat N 60x/1.40 silicon oil objective or an Axio Observer Zeiss microscope (Zeiss, Oberkochen, Germany) with $z$-stacks. Co-localization studies were performed using the ImageJ co-localization plugin JACoP.

### CI-MPR uptake and trafficking analysis

Cells were incubated with 10 µg/ml mouse anti-CI-MPR monoclonal antibody (ThermoFisher MA1-006) in serum-free DMEM for 1 h, rinsed with PBS, fixed with 4% paraformaldhyde, permeabilized with Triton-X100 and immunostained with 1:500 anti-Giantin (rabbit) antibody (BioLegend cat 924302) and then fluorescent anti-rabbit and anti-mouse antibodies 1:500 as described above. A 10-µm area around the Golgi was selected and the co-localization between CI-M6PR and giantin was measured using JACoP plugin in Fiji.

### Live-cell imaging in mammalian cells

For live imaging, cells were plated in an 4-well chambered coverglass (Ibidi µ-slide; Ibidi GmbH, Germany) and medium was replaced with warm imaging buffer (5 mM dextrose (D( + )-glucose), 1 mM $CaCl_2$, 2.7 mM KCl, 0.5 mM $MgCl_2$ in PBS) just before imaging. Images were taken at 37 °C on an inverted Axio Observer Zeiss microscope using a Plan Apochromat N 63x/1.40 oil DIC M27 objective with a Photometrics Prime 95B camera. $z$-stack images were processed using the OMERO client server web tool and Fiji. In Supplementary Fig. 4e, HeLa cells were fixed with 1% ice cold PFA for 15 min and immediately imaged under the microscope to preserve the fluorescence and to stabilize the structure. For kiss-and-run studies, live movies were taken at 37 °C by using imaging buffer on a DeltaVision OMX Optical Microscope equipped with $CO_2$ supply. A 60x/1.524 oil objective was used for image acquisition. 400 nm of thickness was imaged in three $z$-stacks (200 nM each $z$-stack, with oversampling). Movies were further processed using the deconvolution software- SoftWoRx 7.0.0 release RC

6, from Cytiva (now part of Leica). According to the Meta data, the frame rate was 750–780 ms.

### Quantification of Rab11-positive endosomes

Segmentation and analysis were performed on manually chosen ROIs using a custom script for Fiji57 as follows. First, a 3D white top-hat filter58 was applied to the original image to homogenize the background and used to compute 3D seeds59 with subpixel accuracy. Next, objects were segmented on the original image using an iterative threshold60 and converted to labels. Touching objects were then separated by a 3D watershed61 using the previously identified seeds on the label image. The resulting image was then added to the 3D ROI Manager10 to exclude remaining laterally touching objects and finally perform intensity and size measurements per object. A total of 2500–3000 Rab11-positive particles were analyzed from 40 to 50 cells for each condition. The script is available at https://doi.org/10.5281/zenodo.6907356.

### Antibodies and Halo Ligand

The following antibodies were used in this study: Polyclonal rabbit anti-GFP (TP401; Torrey Pines; 1:2,000 for western blotting and 1:200 for immunostaining), anti-Rab10 (D36C4) Rabbit mAb (#8127, Cell signaling, 1:1000 for western blotting), anti-Rab11 antibody (ab3612, abcam, 1:1000 for western blotting), Recombinant anti-SNX5 antibody (ab180520, abcam, 1:1000 for western blotting). Purified Mouse anti-SNX1 (611482, BD biosciences, 1:100 for IF), anti-SNX6 mouse monoclonal antibody (D-5) (sc-365965, Santa Cruz, 1:1000 for western blotting), For pulldowns, Trap beads (nanobodies) were used. GFP-Trap_A (chromotek, gta-20) was used for GFP pulldowns. HRP-conjugated goat anti-mouse IgG (H + L) secondary antibody (ThermoFisher Scientific; 31430; 1:10,000) and polyclonal HRP-conjugated goat anti-rabbit IgG (ThermoFisher Scientific; 31460; 1:10,000) were used (incubated for 1 h at room temperature) to detect bound antibodies with Blotting detection kit WesternBright™ ECL (advansta; K-12045-D50). Alexa Fluor 488–goat anti-rabbit IgG (H + L) (Invitrogen; A-11034) and Alexa Fluor 594–goat anti-mouse IgG (H + L) cross-adsorbed secondary antibodies (Invitrogen; R37121) were used for immunofluorescence. Janelia Fluor® 646 HaloTag® Ligand (Promega; Catalog number GA1120) was used for cargo transfer experiments.

### DNA and plasmid sources

The following commercially available plasmids were obtained: GFP-VIPAS39 (Sino.Bio; HG22032_ACG), GFP-Rabenosyn-5 (Addgene; 37538), turbo-GFP-SNX1 (Origene; RG201844), GFP-RAB11 (Addgene; 12674) sigma1-EGFP (Addgene; 53611). The plasmids Px458 GFP (Addgene; 48138) and Px459 Puro (Addgene; 62988) were used for cloning gRNAs. GFP-VPS16 and TfR-Halo were a kind gift from Paul Luzio and Matthew Kennedy, respectively. mCherry-SNX1 was created by PCR amplifying mCherry from a mCherry-

Rab11 construct (Plasmid #55124) and ligation (Roche rapid ligation kit) into the NotI/PmeI sites of the SNX1 vector (Origene; RG201844) from which Turbo-GFP was deleted by restriction digestion (NotI/PmeI). mCherry was cloned into EGFP-RAB10 vector (Addgene; 49472) after deleting EGFP. mCherry was synthesized from a mCherry Rab11 plasmid (Addgene; 55124).

### Reporting summary
Further information on research design is available in the Nature Research Reporting Summary linked to this article.

## Data availability
The data generated in this study are provided in the Supplementary Information/Source Data file. Source data are provided with this paper.

## Code availability
The script for the quantification of Rab11-positive vesicles is available at https://doi.org/10.5281/zenodo.6907356.

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

## Acknowledgements
We thank Mirjam Pennauer (Biozentrum) and Shahidul Alam (Pharmazentrum) for sharing the Px458 mCherry plasmid and the pUC19 plasmid, respectively. The Halo-tagged TfR and the GFP-VPS16 constructs were a kind gift of Matthew Kennedy (University of Colorado, Denver, Co, USA) and Paul Luzio (CIMR, Cambridge UK), respectively. The Imaging Core Facility of the Biozentrum facilitated the generation of movies and the image analysis, in particular Alexia Loynton-Ferrand and Laurent Guerard. Cells were sorted in the FACS Core Facility of the Biozentrum. We are grateful to Jean Gruenberg for critical comments on the manuscript. Barth Grant, Pingsheng Liu and Alicia Melendez are acknowledged for sharing strains. Some strains were provided by the CGC, which is funded by the NIH Office for Research Infrastructure Programs

(P40 OG010440). This work was supported by the Swiss National Science Foundation (CRSII3_141956, 31003A_141207, 310030_197779 to A.S) and the University of Basel.

## Author contributions

Conceptualization: J.A.S., H.R., and A.S. Methodology: J.A.S., H.R., and A.S. J.A.S. and H.R. performed experiments, analyzed data and prepared the figures. A.S. and J.A.S. wrote and edited the manuscript. H.R. commented on the manuscript. A.S. supervised the work.

## Competing interests

The authors declare no competing interests.
