## [Peer Review File · Nature Communications]

FERARI and cargo adaptors coordinate cargo flow through sorting endosomesEditorial Note: This manuscript has been previously reviewed at another journal that is not operating a transparent peer review scheme. This document only contains reviewer comments and rebuttal letters for versions considered at *Nature Communications*.

REVIEWERS' COMMENTS

Reviewer #1 (Remarks to the Author):

The authors describe a novel role of the FERARI complex in functioning in fusion and fission, allowing for cargo flux between Rab5 and Rab11 cargoes. This intriguing concept would explain why a complex has both components identified to promote fusion (Vps45) and fission (EHD1). To reveal the role of FERARI components, the authors traced selected components relative to cargo and endosomal proteins and indeed observe periodical contacting events, which have not been seen to this degree in the past. Their manipulation of cargo amounts suggest that this periodicity is linked to function.

There is some challenge in the manuscript due to the multiple panels, two parallel systems (mammalian cells, worms), time lapse imaging and data analysis. This is also mainly a cell biology study, while in vitro work is not included - and issue for future studies in my view.

Apparently, the data analysis was vastly improved during the first revision and further improvements were done beyond - which convinced one, but not all reviewers. In my view, the authors addressed, however, all points of reviewer 1, and provide solid answers to reviewer 3, including additional quantification and convincing arguments on why they used the available GFP-lines for AP-1 or SNX proteins.

I therefore feel that the study belongs into the journal and deserves publication in its current form. It is also an intriguing example of a novel function of coupling fusion to fission to sort cargo. This concept may explain how cargo sorting is achieved at endosomes and possibly similarly at the ER-Golgi interface.

Reviewer #2 (Remarks to the Author):

I was asked to provide an opinion on this manuscript, not as a conventional reviewer, but rather as a referee of the unresolved disagreements between the authors and Reviewers 1 and 3. As I see it, there are two general issues.

The first issue is presentation. Apparently the original manuscript was confusing and full of errors in figure labeling, etc. According to the reviewers, these problems have been partially but not entirely corrected in the revised manuscript. Having seen only the revised manuscript, I can say that it is still hard to follow. The logic of the experiments and arguments is often unclear, and I had difficulty absorbing the data and seeing the effects that the authors describe. In other words, the responses of the reviewers are understandable. Looking ahead, the authors would benefit from efforts to present their findings in a more accessible way.

The second issue has to do with the overall picture of the endosomal system. As an outsider, I find the endosomes field to be extraordinarily murky. The relationships between the variously named endosomes seem to be poorly understood, and different researchers describe the system in different ways. Some of those divergent opinions are evident in the discussion here. Given this situation, it seems reasonable that the authors should have leeway to present their interpretations in a way that makes sense to them while acknowledging relevant literature and opposing viewpoints.

My recommendation is that the authors should make one more round of minor revisions to address remaining concerns from the reviewers and to make the story flow as smoothly as possible, using their judgment about which changes are important and justified. During this process, the authors should try to rein in their frustration, because the reviewers have made a good faith attempt to provide constructive criticism. The work will then be in a form that merits being published so that it can be evaluated by the research community.

We wish to thank the reviewers for their positive assessment of our manuscript!

Reviewer #1 (Remarks to the Author):

The authors describe a novel role of the FERARI complex in functioning in fusion and fission, allowing for cargo flux between Rab5 and Rab11 cargoes. This intriguing concept would explain why a complex has both components identified to promote fusion (Vps45) and fission (EHD1). To reveal the role of FERARI components, the authors traced selected components relative to cargo and endosomal proteins and indeed observe periodical contacting events, which have not been seen to this degree in the past. Their manipulation of cargo amounts suggest that this periodicity is linked to function.

There is some challenge in the manuscript due to the multiple panels, two parallel systems (mammalian cells, worms), time lapse imaging and data analysis. This is also mainly a cell biology study, while in vitro work is not included - and issue for future studies in my view.

We agree with the reviewer that the manuscript is rather complex. Unfortunately, in the past we were often confronted with comments from reviewers that what we find in *C. elegans* might not be transferable to mammalian cells. To show evolutionary conservation, we perform experiments in *C. elegans* and mammalian cells.

We also agree with the reviewer that the next step is more in vitro work and biochemical reconstitution of the processes, and we are working towards this goal.

Apparently, the data analysis was vastly improved during the first revision and further improvements were done beyond - which convinced one, but not all reviewers. In my view, the authors addressed, however, all points of reviewer 1, and provide solid answers to reviewer 3, including additional quantification and convincing arguments on why they used the available GFP-lines for AP-1 or SNX proteins.

Thank you very much for your positive assessment!

I therefore feel that the study belongs into the journal and deserves publication in its current form. It is also an intriguing example of a novel function of coupling fusion to fission to sort cargo. This concept may explain how cargo sorting is achieved at endosomes and possibly similarly at the ER-Golgi interface.

Thank you !!!

Reviewer #2 (Remarks to the Author):

I was asked to provide an opinion on this manuscript, not as a conventional reviewer, but rather as a referee of the unresolved disagreements between the authors and Reviewers 1 and 3. As I see it, there are two general issues.

The first issue is presentation. Apparently the original manuscript was confusing and full of errors in figure labeling, etc. According to the reviewers, these problems have been partially

but not entirely corrected in the revised manuscript. Having seen only the revised manuscript, I can say that it is still hard to follow. The logic of the experiments and arguments is often unclear, and I had difficulty absorbing the data and seeing the effects that the authors describe. In other words, the responses of the reviewers are understandable. Looking ahead, the authors would benefit from efforts to present their findings in a more accessible way.

We edited the manuscript to improve the flow and to provide better a rational for the experiments. In addition, we removed some typos and modified parts to fit Nature Communication publication style. We agree that the manuscript is very complex, and in hindsight, it might have been better to have split it into two. However, we wanted to give an as complete picture as possible, and not using the-smallest-publishable-unit approach. The extensive previous revisions also made the manuscript more complex. Notwithstanding, the previous revision improved the manuscript, in particular the tour-de-force experiment showing direct cargo transfer.

The second issue has to do with the overall picture of the endosomal system. As an outsider, I find the endosomes field to be extraordinarily murky. The relationships between the variously named endosomes seem to be poorly understood, and different researchers describe the system in different ways. Some of those divergent opinions are evident in the discussion here. Given this situation, it seems reasonable that the authors should have leeway to present their interpretations in a way that makes sense to them while acknowledging relevant literature and opposing viewpoints.

We added a paragraph in the discussion to address this point.

My recommendation is that the authors should make one more round of minor revisions to address remaining concerns from the reviewers and to make the story flow as smoothly as possible, using their judgment about which changes are important and justified. During this process, the authors should try to rein in their frustration, because the reviewers have made a good faith attempt to provide constructive criticism. The work will then be in a form that merits being published so that it can be evaluated by the research community.

As outlined above, we modified the manuscript. We wish to thank the reviewer for his/her positive assessment of our work.